# Cross-Country Skiing Analysis and Ski Technique Detection by High-Precision Kinematic Global Navigation Satellite System

**DOI:** 10.3390/s19224947

**Published:** 2019-11-13

**Authors:** Masaki Takeda, Naoto Miyamoto, Takaaki Endo, Olli Ohtonen, Stefan Lindinger, Vesa Linnamo, Thomas Stöggl

**Affiliations:** 1Faculty of Health and Sports Science, Doshisha University, Kyoto 610-0332, Japan; parukoz00@gmail.com; 2New Industry Creation Hatchery Center, Tohoku University, Sendai 980-8576, Japan; miyamoto@tohoku.ac.jp; 3Faculty of Sport and Health Sciences, University of Jyväskylä, FI-40014 Jyväskylä, Finland; olli.ohtonen@jyu.fi (O.O.); vesa.linnamo@jyu.fi (V.L.); 4Department of Food and Nutrition and Sport Science, University of Gothenburg, SE-405 Gothenburg, Sweden; stefan.lindinger@gu.se; 5Department of Sport and Exercise Science, University of Salzburg, 5020 Salzburg, Austria; thomas.stoeggl@sbg.ac.at

**Keywords:** classical technique, cross-country skiing, kinematic GNSS, GPS

## Abstract

Cross-country skiing (XCS) embraces a broad variety of techniques applied like a gear system according to external conditions, slope topography, and skier-related factors. The continuous detection of applied skiing techniques and cycle characteristics by application of unobtrusive sensor technology can provide useful information to enhance the quality of training and competition. (1) Background: We evaluated the possibility of using a high-precision kinematic global navigation satellite system (GNSS) to detect cross-country skiing classical style technique. (2) Methods: A world-class male XC skier was analyzed during a classical style 5.3-km time trial recorded with a high-precision kinematic GNSS attached to the skier’s head. A video camera was mounted on the lumbar region of the skier to detect the type and number of cycles of each technique used during the entire time trial. Based on the GNSS trajectory, distinct patterns of head displacement (up-down head motion) for each classical technique (e.g., diagonal stride (DIA), double poling (DP), kick double poling (KDP), herringbone (HB), and downhill) were defined. The applied skiing technique, skiing duration, skiing distance, skiing speed, and cycle time within a technique and the number of cycles were visually analyzed using both the GNSS signal and the video data by independent persons. Distinct patterns for each technique were counted by two methods: Head displacement with course inclination and without course inclination (net up-down head motion). (3) Results: Within the time trial, 49.6% (6 min, 46 s) was DP, 18.7% (2 min, 33 s) DIA, 6.1% (50 s) KDP, 3.3% (27 s) HB, and 22.3% (3 min, 03 s) downhill with respect to total skiing time (13 min, 09 s). The %Match for both methods 1 and 2 (net head motion) was high: 99.2% and 102.4%, respectively, for DP; 101.7% and 95.9%, respectively, for DIA; 89.4% and 100.0%, respectively, for KDP; 86.0% and 96.5%, respectively, in HB; and 98.6% and 99.6%, respectively, in total. (4) Conclusions: Based on the results of our study, it is suggested that a high-precision kinematic GNSS can be applied for precise detection of the type of technique, and the number of cycles used, duration, skiing speed, skiing distance, and cycle time for each technique, during a classical style XCS race.

## 1. Introduction

Cross-country skiing (XCS) in the classical style embraces a broad variation of techniques applied like a gear system according to slope inclination, skiing speed, snow conditions, grip, and glide of the skis, and factors related to the skier (e.g., skiing skills, energy efficiency, strength capacities, anthropometrics) [1]. The main techniques, in order of lowest to highest gear, are the herringbone technique (HB), diagonal stride (DIA), kick double poling (KDP), double poling (DP), and crouching (downhill). If the skier can choose a faster technique, it will result in shorter competition time. Therefore, the detection and analysis of technique are important not only for improving athletic performance, but also for researchers and media related to XCS.

Although the most practical approach to quantify the application of ski technique during a race is an analysis of video data, XCS tracks are often long (2.5–10 km laps) and run through forests and mountains, making the use of video recordings impractical in terms of cost and effort. Some studies have analyzed the ski technique application during a race on selected spots of a track using 2D video analysis [2,3,4]. However, there is a research deficit concerning how much each technique is applied during an entire XCS race or time trial. Stöggl et al. [5] used a smartphone internal accelerometer data for the detection of different skating techniques in a laboratory setting, but not during on-snow skiing and competition. Marsland et al. [6] applied a single inertial measurement unit (IMU) during training to develop and validate algorithms for the detection of classical XCS techniques and basic cycle characteristics. This methodology was shown to be valid, with a total %Match (the number of techniques counted by sensor/the number of techniques counted by video) of 83.8%, with the highest values for DIA (87.9%), and lower values for DP (77.5%) and KDP (74.2%). Marsland et al. [7] recently applied this methodology to compare techniques used and cycle characteristics between a simulated distance (10.5 km) and sprint (1.1 km) XCS race. They revealed that the differences for cycle counts were -1.14% ± 30.3% for DP, -12.3% ± 8.3% for DIA, and -28.4% ± 23.8% for KDP, and for technique duration -1.3% ± 28.8% for DP, -10.0% ± 7.5% for DIA and -30.4% ± 21.1% for KDP, comparing the number of cycles of each technique with the video data [6]. The proportion of each technique was also assessed as 43% ± 5% for DP, 5% ± 4% for KDP, and 16% ± 4% for DIA, except for downhill during a classical race [8]. The accuracy of cycle counts of 83.8% revealed from these studies was high but the standard deviations (for example, DP 30.3%, DIA 8.3%, and KDP 23.8%, for cycle counts) were high, even in a measurement section of a short distance, of a flat to moderate uphill, and at a moderate skiing speed [6]. The reasons for less accuracy would depend on the accuracy of sensors, measurement frequency, and specific difficulties of counting the techniques, with small differences in physical and skiing motions among several techniques in an XCS classical style race. Another problem of IMUs could be that they cannot measure skiing velocity during XCS. More accurate skiing analysis and detection methods for the complex motions of XCS are required for athletes, coaches, and researchers.

Andersson et al. [9] used a Leica differential global navigation satellite system (d-GNSS) consisting of (1) Leica GX1230 GG, 72 channel, dual-frequency L1/L2 receivers, (2) Leica AX1202 GG survey antennas, and (3) Leica GFU14 Satellite 3AS radio modems (Leica Geosystems AG, Heerbrugg, Switzerland), with a total weight of 1.67 kg. The main system attached to the skier’s back and the antenna was positioned on the upper back to analyze the skiing velocity during a simulated sprint XCS time trial. However, it was not used to detect ski technique application or cycle characteristics from GNSS data. This system was quite accurate for obtaining position data. However, this system was heavy (1.67 kg) and not realistic for athletes in both sprint and distance ski training and races. In their study, technique distribution and cycle characteristics were calculated based on video data (from a snowmobile following the skier). Bolger et al. [10] used a global positioning system (differential GPS; Garmin Forerunner GPS watch, Garmin Ltd., Olathe, KS) that collected position and heart-rate data at a 1-Hz sampling rate to analyze average skiing speed in uphill, flat, and downhill sections in 15-km (men) and 10-km (women) simulated classic and skating XCS competitions, but without a focus on ski technique application. In this system, sampling frequency and accuracy of positioning data were too low to detect technique in XCS. Therefore, there is no definitively accurate method to detect techniques and speed in XCS to date. 

Miyamoto et al. [11] developed a lightweight, compact, and highly accurate kinematic GNSS logger (AT-H-02, AOBA Technologia LLC). A commonly applied GPS is usually a single point GPS with an accuracy of around 10 m. Relative positioning GPS (differential GPS) is a more accurate GPS method than a single point GPS. Although the positioning accuracy can be improved using the correction information obtained from the pseudo-distance from the GPS satellite observed at a known base station, its accuracy is still approximately 1 m. The kinematic GNSS logger developed by Miyamoto et al. [11] applies a configuration specialized to the post-processing kinematic method without using any real-time processing, such as the navigation possessed by an ordinary GNSS. Since the kinematic method uses the phase information of the GNSS satellite carrier, the positioning error is reduced to a 1.0–2.0 cm level (sampling frequency 10 Hz) [12]. Therefore, this system currently demonstrates the highest precision of commercially available GPS. The question remains, therefore, whether higher precision GNSS might result in higher detection rates for all classical style XCS techniques and, in particular, whether all techniques in the cross-country skiing classic style have distinctly different up and down head motions. If the position change of the skier’s head can be detected with a high-precision GNSS at the centimeter scale, it can be used for technique detection and analysis of classical cross-country skiing. Thus, the aim of the current study was to analyze if the kinematic GNSS data obtained from the head displacement of an XC skier is feasible and valid for detection of the number of skiing cycles, the applied skiing technique, and the basic cycle characteristics during a XCS simulated race.

## 2. Materials and Methods

One world-class male XCS athlete was asked to perform a classical style time trial on a F.I.S. (International Ski Federation) homologated XCS course of 5.3 km (two laps of 2.65 km each, Figure 1) [13]. The ski course was located at Vuokatti, Finland. Figure 1 shows the three-dimensional mapping data of the cross-country ski course used in this study in Google Earth Pro. Plotted data on this figure were obtained from the actual experiment of this study subject (one lap: 2.65 km). Glide and grip waxes were arranged by a professional serviceman of the Finish national XCS team. A small and high-precision kinematic GNSS (AT - H - 02, AOBA Technologia LLC.; size: 78 × 38 × 18 mm; weight: 69 g; sampling rate: 10 Hz) equipped with an external antenna was mounted on the head. The measurement accuracy of the GNSS used was shown to be 4.58 cm in altitude by static positioning analysis [12]. In this study, we found the altitude measurement accuracy on the XCS course was 1.34 cm at the minimum and 5.66 cm on average. Moreover, altitude change was much more accurate than the altitude when the same satellite combination was used for positioning calculations. Therefore, we were able to obtain clear up-down head motions using the high-precision GNSS. There were trees in the ski course in some places. However, the width of the ski course was approximately 20 m and the sky was open above the full course. The experiment was conducted on snow, during March, when tree branches were not as thick as those shown in Figure 1. The reference station was placed on a fixed tripod with open sky to assure maximum accuracy and ease of measurement of the GNSS. The study received approval from the local Ethical Committee (EK-GZ: 05/2017) and was conducted in accordance with the Declaration of Helsinki.

In the first step, the trajectory of the motion of the head (head vertical displacement) with respect to latitude, longitude, and altitude were obtained from the raw data of the GNSS with post-processing kinematic software, rtklib [4], for the entire XCS time trial. Head vertical displacement included the slope inclination of the XCS course.

In the second step, to achieve the slope inclination across the entire XCS course, a 0.9-second (s) moving average was applied to the altitude data obtained from the raw data of the GNSS using rtklib [4]. The reason for calculating the moving average every 0.9 s was as follows. In order to show the change in the inclination angle of the course smoothly, it was reasonable to calculate a moving average of as short a time interval as possible. However, the moving average of every 0.5 s was considered to have too few plotted data. As a result of various trials, 0.9 s was found to be the most suitable. Note, however, that there was no deep physical or mathematical meaning in the chosen averaging period of 0.9 s, and that 0.8 or 1.0 s may be more appropriate depending on the course. In this study, it was necessary to perform technique detection and analysis using only a lightweight and compact GNSS measuring device. Thus, the averaging period was set to about 0.9 s. With this procedure, it was possible to draw the smoothed slope inclination (altitude) change for the entire XCS course (course profile) as shown in Figure 2.

In the final step, the change in the trajectory of the net vertical displacement of the head was extracted by subtracting the slope inclination data from the altitude data obtained from the raw data of the GNSS. These two trajectory patterns of vertical displacement were used to define waveforms representative of the techniques DP, DIA, KDP, HB, and downhill skiing. The typical waveform patterns for each technique were obtained, respectively, as shown in Figure 3 for DP, Figure 4 for DIA, Figure 5 for KDP, and Figure 6 for HB. In these figures, the upper panel illustrates the raw waveform of the head displacement, including the slope inclination. The middle panel shows the slope inclination calculated based on the 0.9 s moving average of the head displacement, and the bottom panel illustrates the extracted net head displacement (up-down motion) excluding slope inclination.

To obtain external criteria as a reference, a video camera (Go Pro Hero 5, 50 Hz) was attached to the lower back of the participant with a total view of both skis and poles. This data was visually analyzed to determine the type of technique and the number of cycles using the Kinovea video software. All techniques in the video image were clearly visualized. The waveform pattern of vertical displacement of the head from the GNSS data for each technique was clarified by matching with the video image. For validation purposes, the number of cycles for each technique measured from both GNSS trajectory data with slope inclination (first step) and net vertical up-down head motion (without slope inclination) was compared with those obtained from the video data based on the 1-min successive example analysis as shown in Figure 2. Here, the head vertical displacement is shown in Figure 7. Net vertical displacement, slope inclination (0.9 s moving average of head displacement), and skiing speed change are shown in Figure 8. After confirmation of the data, the number of cycles were counted manually (visually) and cycle time (CT) was calculated for each technique throughout the whole-time trial according to GNSS-based waveforms of head and net vertical displacement. One waveform cycle corresponded to one skiing cycle. Counting of the number of cycles based on waveforms (head and net vertical displacement) and video images was conducted by three different persons. %Match (%) was defined and calculated as the ratio of the counts from the GNSS data to the counts from the video data as the reference (%Match = GNSS data/Video data). Absolute time trial duration, relative time using each technique as a percentage of total time, absolute skiing distance, relative skiing distance as a percentage of total distance, average skiing speed, and cycle time for DP, DIA, KDP, HB, and downhill were measured and analyzed from the GNSS data. All data are presented as mean values and standard deviations.

## 3. Results

### 3.1. Technique Waveform Pattern for Each Technique

In both Figure 7 and Figure 8, the patterns of waveforms according to each skiing technique and cycle durations are represented. Within this section, the participant used 6 DPs, 6 KDP, 6 DPs, 34 DIAs, 1 KDP, 7 DPs, 10 HBs, 2 DIAs, and downhill according to the course terrain. Average CTs were 1.1 s for DP, 0.5 s for DIA, 1.3 s for KDP, and 0.4 s for HB. Within the 1-min section for waveform definition, for DP (Figure 3), the change in vertical head displacement was 48.8 ± 2.2 cm with a cycle time of 1.10 ± 0.07 s on average. For DIA (Figure 4), the vertical head displacement was 19.8 ± 0.9 cm (approximately 2.5 times lower than that of DP) with a cycle time of 0.60 ± 0.06 s (54% of that of DP). KDP (Figure 5) was characterized by a double peak within each cycle. The first maxima in head displacement was prior to pole plant, followed by a lowering during the first part of the poling phase (trunk, knee, hip flexion), and an increase towards the second maxima at the end of the passive backward swing phase and prior to the start of the leg push-off. Head displacement of KDP was 29.5 ± 6.8 cm (60% of that of DP) with a cycle time of 1.26 ± 0.15 s (15% longer than that of DP). For HB (Figure 6), the head movement demonstrated the smallest displacement of approximately 10.0 ± 2.7 cm (5 times lower than that of DP) with the shortest cycle time of 0.44 ± 0.06 s (40% of that of DP).

For downhill skiing, unlike in the other techniques, the vertical head displacement was more conducive to detection based on the non-cyclic vertical displacement of the head motion (Figure 7 and Figure 8).

### 3.2. Skiing Characteristics and Technique Distribution during the Time Trial

The skiing time for the 5.3 km trial was 13 min, 39 s. Within each technique, the absolute skiing time, relative skiing time, absolute distance and relative distance were, respectively: 6 min, 46 s (49.6%) and 2,741 m (51.4 %) for DP; 2 min, 33 s (18.7%) and 216 m (4.0 %) for DIA; 50 s (6.1%) and 573 m (10.7 %) for KDP; 27 s (6.1%) and 64 m (1.2%) for HB; and 3 min, 03 s, (22.3%) and1,743 m (32.7%) (Table 1). Skiing velocity and cycle time were, respectively; 21.1 ± 2.9 km/h and 1.10 ± 0.07 s for DP; 14.5 ± 1.7 km/h and 0.60 ± 0.06 s for DIA; 15.1 ± 1.2 km/h and 1.26 ± 0.15 s for KDP; 9.2 ± 1.4 km/h and 0.44 ± 0.06 s for HB; and 35.3 ± 5.8 km/h for downhill. 

The distribution of the five techniques during the time trial is shown in Figure 9, Figure 10 and Figure 11. Figure 9 indicates GNSS data plotted east–west (*X*-axis) and north–south (*Y*-axis). Each technique was identified and described by the net vertical displacement of the skier’s head. Figure 10 and Figure 11 also indicate the technique distribution with altitude (Figure 10) and horizontal velocity (Figure 11) of the skier’s head plotted on the vertical axis. 

The %Match based on GNSS counts with altitude for each technique were 99.2% for DP, 101.7% for DIA, 89.4 % for KDP, and 86.0% for HB (Table 2). Total %Match, including all four techniques, was 98.6%. On the other hand, the %Match based on GNSS counts without altitude for each technique were 102.4% for DP, 95.9% for DIA, 100.0% for KDP, and 96.5% for HB, respectively (Table 2). The average %Match for all four skiing techniques was 99.6%.

## 4. Discussion

Results show that an excellent %Match of 98.2% was obtained in this study. Marsland et al. [6,7,8,14] analyzed classical ski techniques using microsensor units (three-axis accelerometer, gyroscope, GPS sensor) and magnetometer. They revealed that the differences for cycle counts were -1.14% ± 30.3% for DP, -12.3% ± 8.3% for DIA, and -28.4% ± 23.8% for KDP, and for technique duration -1.3% ± 28.8% for DP, -10.0% ± 7.5% for DIA, and -30.4% ± 21.1% for KDP, comparing the number of cycles of each technique with the video data [6]. The proportion of each technique was also detected as 43% ± 5% for DP, 5% ± 4% for KDP, and 16% ± 4% for DIA, except for downhill during a classical race [8]. In our study, the %Match were 99.5% for DP, 100.7% for DIA, 86.0% for KDP, and 86.0% for HB. As a whole, an extremely high average total %Match of 98.2% was obtained. However, future studies should consider analyzing multiple participants using the present methodology to provide information about the mean and standard error of %Match. 

The proportions of each technique used during the entire time trial were 49.6% for DP (6 min, 46 s), 18.7% (2 min, 33 s) for DIA, 6.1% (50 s) for KDP, 3.3% (27 s) for HB, and 22.3% (3 min, 03 s) for downhill. The high proportion of DP use (almost 50%) relative to total skiing time and covered distance demonstrates that modern XCS relies heavily on upper body work [3,5,15,16,17]. The subject of this study had a high DP capacity, being one of the world’s leading skiers in this discipline. Skiers with high DP capacity are able to provide power to the ski pole efficiently, and display effective coupling of the pole and snow, such as in the efficiency and directionality of the power conveyed to the pole [15,17,18,19]. In addition, the ski track and ski waxing were perfectly prepared. These factors made it possible to use a high ratio of DP during the time trial. We did not compare the duration of each technique among different performance levels of skiers. The relationship between the proportions of each technique used and ski performance could be a future application of this study area.

There are other advantages presented by a high-precision GNSS sensor. Within an XCS race, the skier, coaches, or organizers should be able to accurately analyze the skiing speed and time at every part of the course. For athletes and coaches, the target skier could then be compared with other skiers, to analyze tactical and pacing behavior, and section times between skiers. Such information can be used for planning training and developing strategies for the next ski race. Furthermore, this methodology might allow for enriched information for media presentations (e.g., broadcasting of races). In contrast to high-precision kinematic GNSS, the precision of the speed and time evaluation is lower in single point GNSS or differential GPS sensors. The high-precision kinematic GNSS sensor used in this study will enable skiing speed evaluation with considerable accuracy. Although it is possible to detect skiing speed and time using IMUs, data from these devices require considerable post-processing. Furthermore, while an accelerometer cannot remove the effects of the slope inclination of the ski course, a high-precision kinematic GNSS can remove the slope inclination from head displacement and obtain important head positions. In this study, by accurately measuring and extracting the displacement of the head, skiing analysis and technical detection were possible. By fusing GNSS and IMU data, it may be possible to further improve the accuracy of the skiing analysis and technique detection. As shown in Figure 9, Figure 10 and Figure 11, an advantage of our study method using a high-precision kinematic GNSS was the ability to draw technique distribution in conjunction with position data (Figure 9), altitude change of the skier (Figure 10), and horizontal velocity of the skier (Figure 11) on a full course of a time trial. From these figures, it was clear at a glance what technique was used on which part of the ski course, together with altitude change and skiing speed. This information should be useful for skiers, coaches, and scientists to understand a skier’s skiing properties, including not only physical fitness and skiing technique, but also skiing gliding conditions. The average speed for each technique found from this study was 21.1 ± 2.9 km/h for DP, 14.5 ± 1.7 km/h for DIA, 15.1 ± 1.2 km/h for KDP, 9.2 ± 1.4 km/h for HB, and 35.3 ± 5.8 km/h for downhill. It was found that DP was the fastest technique, followed by KDP, DIA, and HB, except for downhill. Additionally, DP was the longest covered distance, followed by KDP, DIA, and HB, except for downhill. Each technique was selected by the skier depending on the skiing speed, the inclination of the slope, the snow, and the ski grip conditions. 

In this research, we extracted vertical displacement of the skier’s head from the obtained GNSS data and counted each technique by visual observation, comparing it with video data as a reference. A disadvantage of this study was the use of only one skier. We have recognized this study limitation. However, we have proved that a small, lightweight, and high-precision kinematic GNSS can supply useful information related to skiing characteristics for skiers, coaches, and scientists in an XCS classical race. In the future, an algorithm-based automatic analysis of skiing properties and technique detection based on kinematic GNSS data should be developed. 

## 5. Conclusions

To detect the type, number of cycles, duration and distance of each technique used during a 5.3 km classical style XCS race, a high-precision kinematic GNSS sensor was attached to the upper back of a skier’s head. The type, number, and cycle duration of each technique based on the motion of the head and net vertical displacement were derived from the GNSS position data. The number of cycles in each ski technique was compared with video data as a reference and %Match was calculated. High total %Match of 98.6% (head vertical displacement) and 99.6% (net vertical displacement) in all techniques was obtained. It was also revealed that almost 50% of skiing duration and total distance was performed with DP, which was the highest speed technique in flat to uphill sections. From the results of this study, it was concluded that a highly accurate kinematic GNSS sensor can satisfactorily detect (1) the type of each technique, (2) the number of cycles for each technique, (3) the skiing duration, (4) the distance covered, (5) skiing speed, and (6) cycle time of each technique, at every part of a time trial during an XCS classical race.

## Figures and Tables

**Figure 1 sensors-19-04947-f001:**
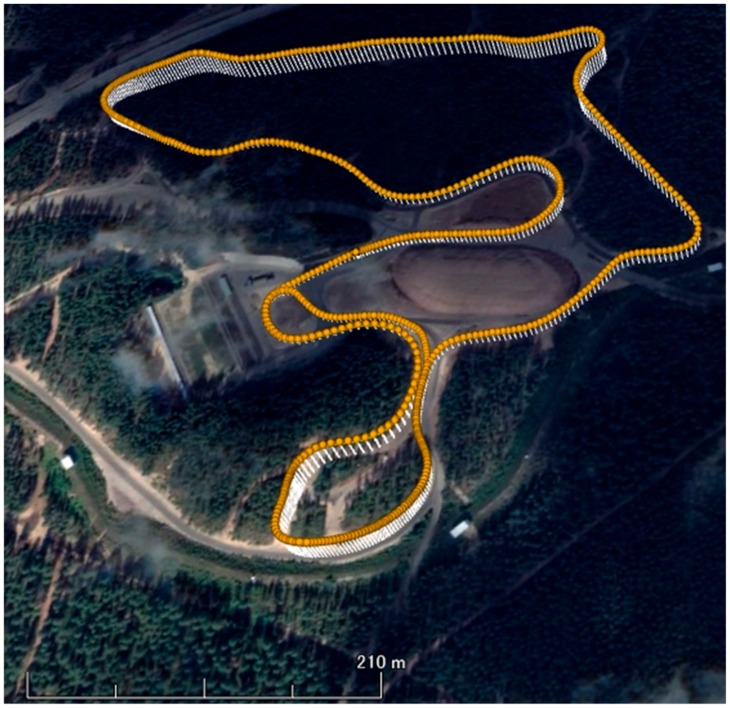
Three dimensional mapping of cross-country ski course, Vuokatti, Finland, used in this study on Google Earth Pro. Plotted data on figure was obtained from the subject of this study (1 lap, 2.65 km). The experiment was conducted on snow, March. The tree branches were not so thick as seen in this picture.

**Figure 2 sensors-19-04947-f002:**
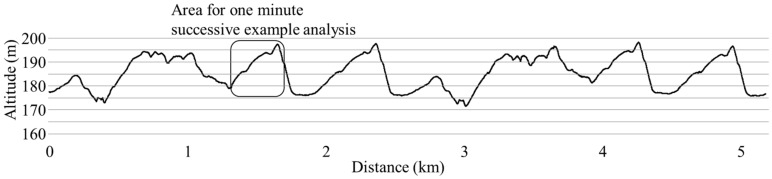
Course profile in this experiment measured by used kinematic GNSS position data. Boxed area was used for one minute example successive technique detection.

**Figure 3 sensors-19-04947-f003:**
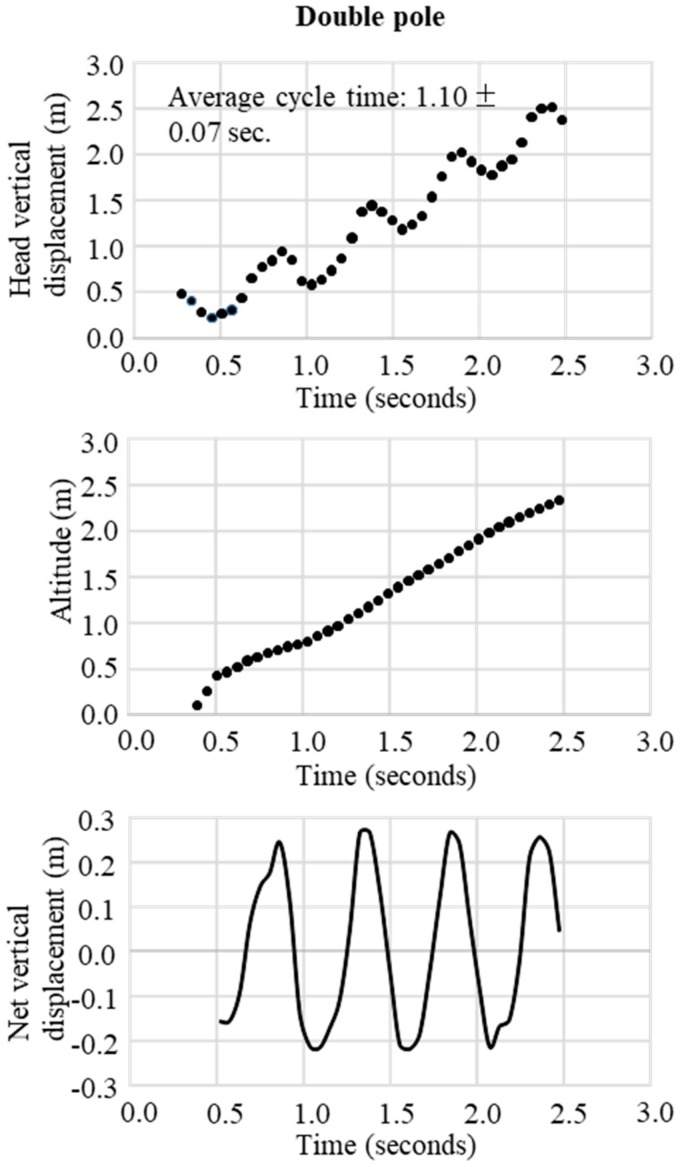
An example of successive double poling obtained from GNSS data. Upper figure is the raw waveform of head vertical displacement obtained from GNSS data, the middle is the change in head altitude (slope inclination) calculated from the 0.9 seconds moving average of head vertical displacement, and the bottom is the net vertical displacement calculated from vertical head displacement–head altitude.

**Figure 4 sensors-19-04947-f004:**
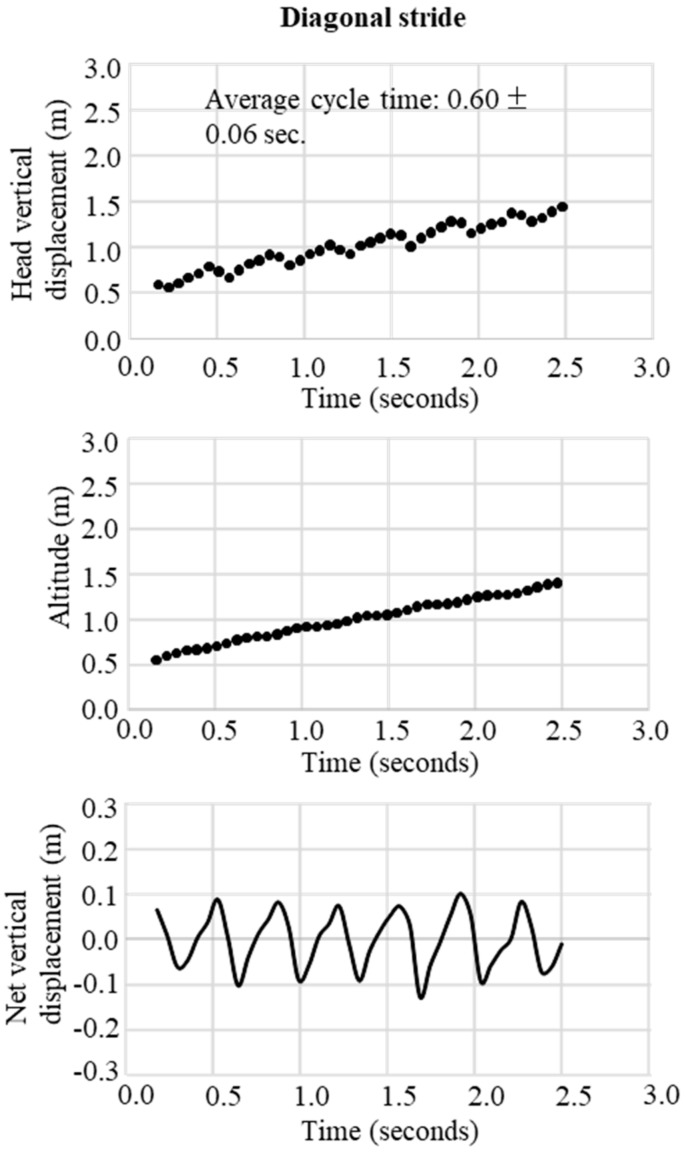
An example of successive diagonal stride obtained from GNSS data. Upper figure is the raw waveform of head vertical displacement obtained from GNSS data, the middle is the change in head altitude (slope inclination) calculated from the 0.9 seconds moving average of head vertical displacement, and the bottom is the net vertical displacement calculated from vertical head displacement–head altitude.

**Figure 5 sensors-19-04947-f005:**
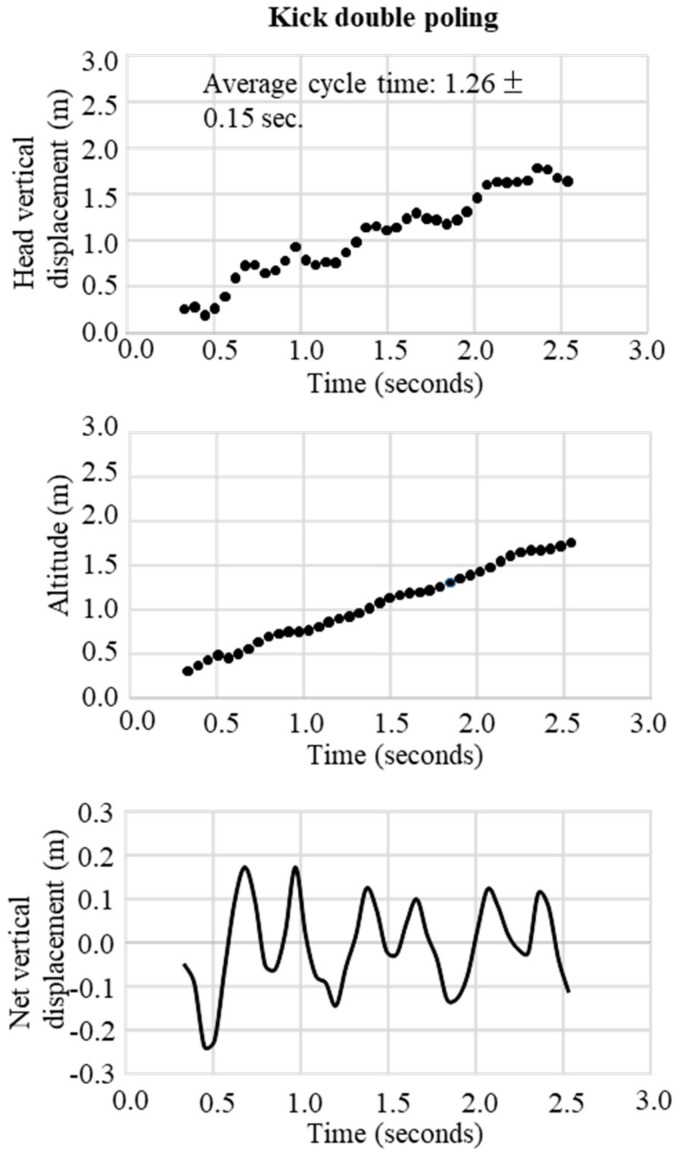
An example of successive kick double poling obtained from GNSS data. Upper figure is the raw waveform of head vertical displacement obtained from GNSS data, the middle is the change in head altitude (slope inclination) calculated from the 0.9 seconds moving average of head vertical displacement, and the bottom is the net vertical displacement calculated from vertical head displacement–head altitude.

**Figure 6 sensors-19-04947-f006:**
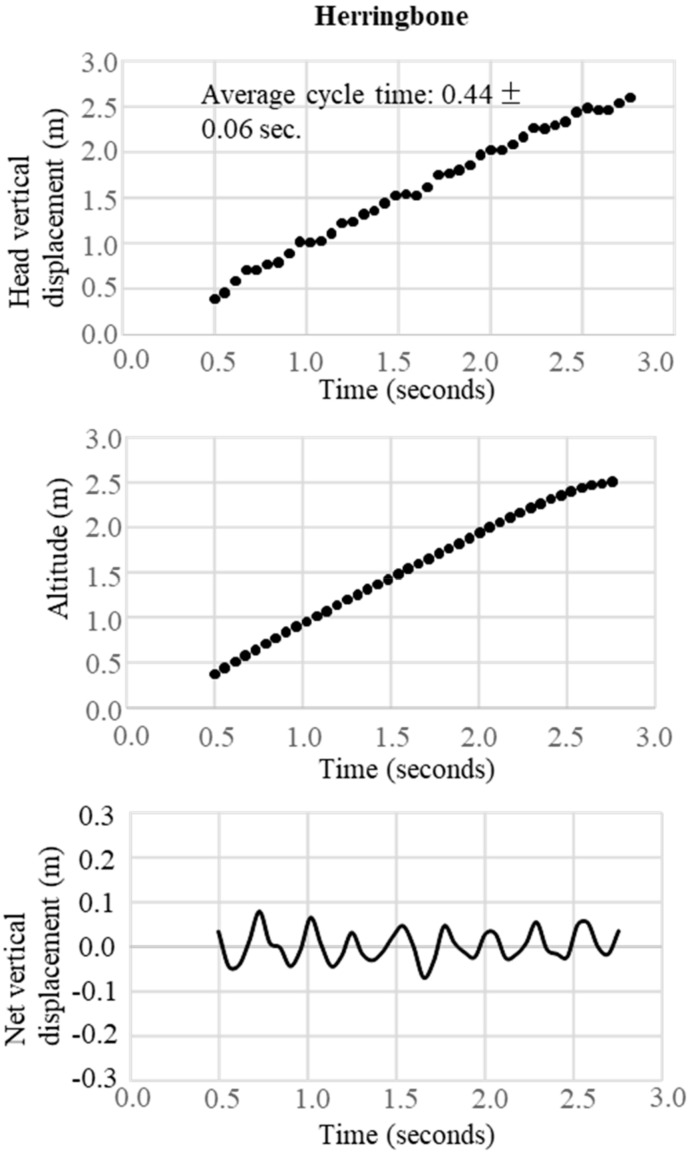
An example of successive herringbone obtained from GNSS data. Upper figure is the raw waveform of head vertical displacement obtained from GNSS data, the middle is the change in head altitude (slope inclination) calculated from the 0.9 seconds moving average of head vertical displacement, and the bottom is the net vertical displacement calculated from vertical head displacement–head altitude.

**Figure 7 sensors-19-04947-f007:**
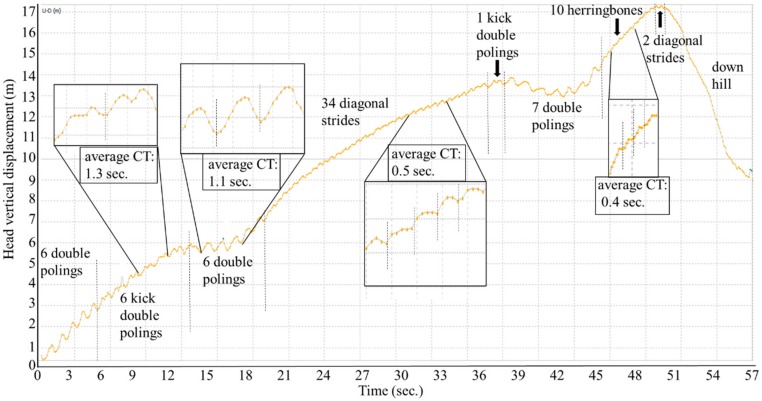
An one minute example of successive change in head vertical displacement accompanying technique changes such as double poling, kick double poling, diagonal stride, herringbone, and downhill obtained from GNSS data. CT: Cycle time.

**Figure 8 sensors-19-04947-f008:**
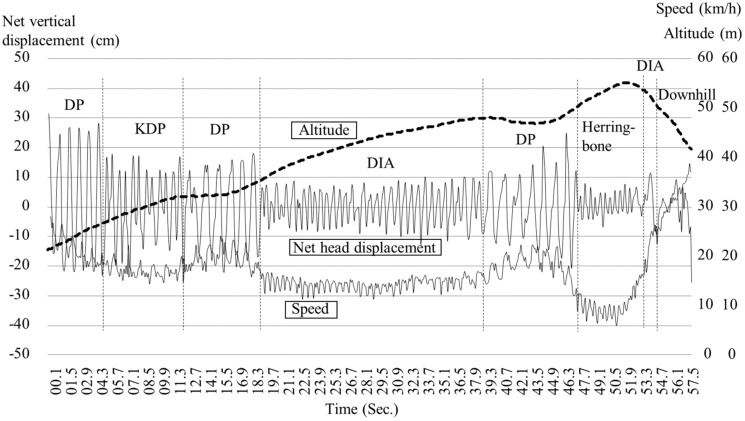
An example of the changes in altitude (slope inclination), net vertical displacement without slope inclination, and skiing speed for successive 1-min analysis.

**Figure 9 sensors-19-04947-f009:**
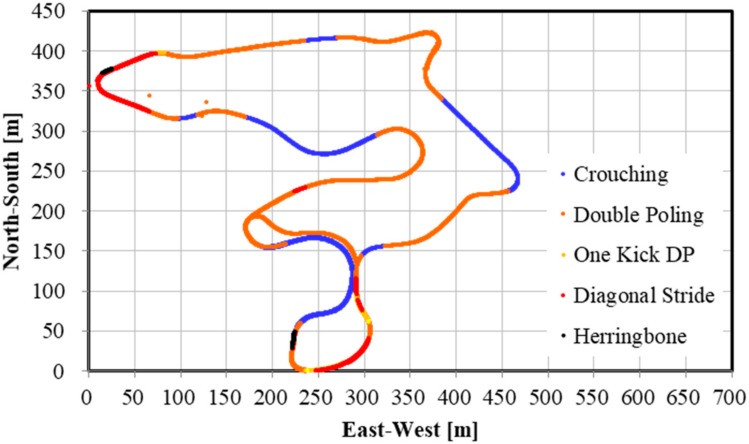
Skiing technique distribution in a 5.3 km XCS classical style time trial. *X*-axis indicates East–West (m) position and *Y*-axis indicates North–South position (m).

**Figure 10 sensors-19-04947-f010:**
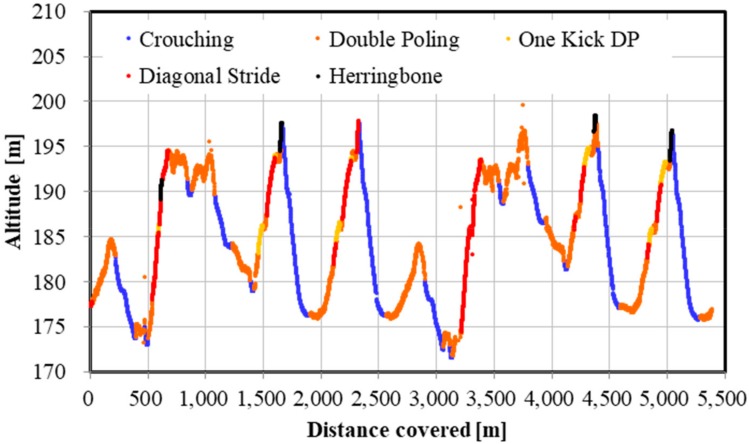
Skiing technique distribution in a 5.3 km XCS classical style time trial. *X*-axis indicates the distance covered and *Y*–axis indicates the altitude of skier’s head.

**Figure 11 sensors-19-04947-f011:**
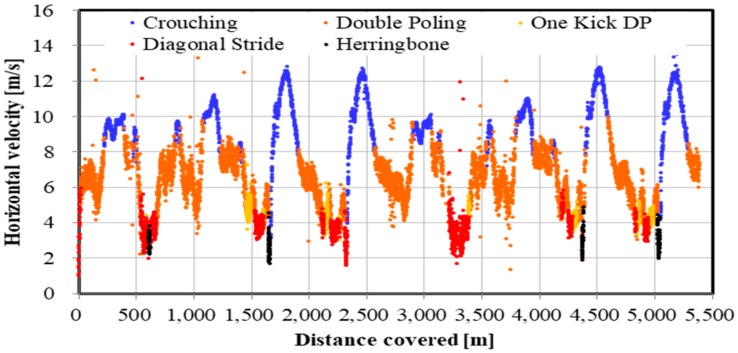
Skiing technique distribution in a 5.3 km XCS classical style time trial. *X*-axis indicates the distance covered and *Y*–axis indicates horizontal velocity of skier’s head.

**Table 1 sensors-19-04947-t001:** Characteristics of skiing, i.e., absolute time, relative time, absolute distance, relative distance, average speed and cycle time covered by each technique.

Technique	Absolute Time (sec)	Relative Time (%)	Absolute Distance (m)	Relative Distance (%)	Average Speed (km/h)	Cycle Time (sec)
**Double poling**	6′46′′	49.6	2740.6	51.4	21.1 ± 2.9	1.10 ± 0.07
**Diagonal**	2′33′′	18.7	216.0	4.0	14.5 ± 1.7	0.60 ± 0.06
**Kick DP**	50′′	6.1	572.5	10.7	15.2 ± 1.2	1.26 ± 0.15
**Herringbone**	27′′	3.3	64.0	1.2	9.2 ± 1.4	0.44 ± 0.06
**Downhill**	3′03′′	22.3	1742.9	32.7	35.3 ± 5.8	
**Total**	13′39′′	100.0	5336.0	100.0	23.2	

**Table 2 sensors-19-04947-t002:** The number of cycles counted by video, kinematic GNSS with altitude (head vertical displacement) and GNSS without altitude (net vertical displacement) and %Match (GNSS/video) in 5.34 km classical ski time trial.

Technique	Video Counts	GNSS Counts with Altitude	%Match (%)	GNSS Counts without Altitude	%Match (%)
**Double poling**	371	368	99.2	377	102.4
**Diagonal**	287	292	101.7′	280	95.9′
**Kick DP**	47	42	89.4	47	100
**Herringbone**	57	49	86	55	96.5
**Total**	762	751	98.6	759	99.6′

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
