# Peer review of "Cross-Country Skiing Analysis and Ski Technique Detection by High-Precision Kinematic Global Navigation Satellite System"

_sensors, 2019, doi:10.3390/s19224947_

Round 1

Reviewer 1 Report

The proposed paper is aligned with the aims and scope of this journal. A technique based on post-processing kinematic GNSS to perform cross-country skiing analysis is presented in this paper. Although the subject is quite interesting, the structure of this research is not fully consistent with the guidelines of Sensors, and the mathematical background, algorithms, and analysis are not thoroughly presented. The bibliography section needs to be augmented and more references should be added. In my opinion, the paper seems short for the standard articles published in Sensors and certain issues must be further seen in greater detail. Overall, it seems more like a conference paper rather than a journal article. The authors are suggested to provide a more detailed version of this manuscript that will elaborate further in the presented aspects of their study. A few comments follow:

Line 80 and 92: The accuracy of 1.0 – 2.0cm refers to the 2D or 3D position? Please support this accuracy statement with additional references. The paper you cite ([10]) cannot be retrieved. Keep in mind that, by default, the vertical component in GNSS suffers in terms of accuracy. Please provide more details regarding the experiment location. Was it a GNSS challenging environment (existence of trees or canopy cover)? For the sake of clarity, it would be good to show a map of the trajectory in Google Earth (or a similar tool) for the reader to have an idea of the experimental conditions. Line 98(i): Elaborate more on how did you come up with the 0.9s moving average window width?  Line 98(ii): The moving average window was applied to the “raw data”. I surmise that you mean the GNSS raw positions? Raw data means pseudorange and phase observables.

Author Response

Thank you for your many suggestions.

I read your comments and it was a very good study for me on writing the scientific article.

According to your comments, my answer and corrections were shown as follows.

Since I am not a native English. Please allow for my poor English.

Your suggestion: The mathematical background, algorithms, and analysis are not fully presented. A bibliographic section needs to be added and a reference needs to be added.

My answer & correction: In this regard, I added text to the introduction (l.63-72).

Your suggestion: About the accuracy of GNSS meter

My answer & correction: The accuracy of 1.0-2.0cm is the accuracy in 3D.

I added a paper to show this, and I deleted it because I couldn't find the paper that I showed before. I will send cited literature.

Your suggestion: About the experiment location

My answer & correction: I added the information about the experiment location in the Method section, and also added the measurement environment (existence of trees, course conditions, etc.).

The location information (raw data) of east, west, south, and north measured with a GNSS device was shown on Google Map and added figure as Fig. 1. Therefore, the No. of other figures were changed according adding new figure.

Your suggestion: The basis for calculating the moving average every 0.9 seconds

My answer & correction: The basis for calculating the moving average every 0.9 seconds is shown in the method section. The 0.9 second moving average is only used to show the course slope change, and the raw data is used to analyze each technique, not the moving average.

In order to show the course profile, for example, measuring GNSS on a snowmobile cannot reproduce the course that the athlete really passed, so it is more reliable to use the data of the GNSS meter attached to the athlete I think.

Reviewer 2 Report

Globally, the manuscript is well written and organized. However, there are some English “bugs” (including missing “.” and “)”) that must be corrected. I recommend a through full read by the authors. The topic under study falls within the scope of the journal.

However, I have some major concerns, which lead me to recommend “major revisions” of the manuscript.

First: the authors have conducted a single experiment (only with one “world-class male XC skier”). I do not believe that we may wish to infer conclusions from a single experiment and state the it will be valid in other situations, or applicable to other users/situations. Even for a non-expert in skiing (like me), it is very obvious that not all the athletes possess exactly the same corporal postures for the same skiing technique; how do these small intra-athletes variations will influence the “mean” of the curves plotted in figures 2,3, etc.? Will the difference between these plots for the different skiing techniques continue to be very obvious?

Second: I do not agree with the results presented in section “3 Results”. In particular, and concerning the values presented in subsection “3.2. Skiing characteristics and match ratio of GNSS based waveforms and video data”, how can they be higher than 100%? Just because they are not correct! Used this way, these values will introduce bias in the final results. If one particular value is incorrect it cannot be used anywhere else, and including using it to produce mean values (the mean value will also be incorrect)!!! You have to treat separately the "correct" and "incorrect" values. To do that you have to: 1st produce a "ground-truth" (for example, using the video as the reference); 2nd measure the percentage of correct and incorrect detections by the GNSS/sensors (relative to that "ground-truth"); 3rd analyze these results separately.

Third: as I consequence, the discussion presented in section 4 and the conclusions in section 5 are biased, and incorrect.

Author Response

Thank you for your many suggestions.

I read your comments and your comments are understandable for me.

Because of my poor explanation in manuscript, I may not have been able to convey what I wanted to say well, so I will explain it supplementarily with the answer.

Since I am not a native English. Please allow for my poor English.

Point 1

This research is the data of the world's top level players with great technique.

Subject was only 1 skier. We are recognizing it as the limitation of our study.

However, we are thinking that this data is very valuable for Sensors for revealing the possibility of high precision kinematic GNSS for analyzing sports activity and in terms of sports science to find out what techniques the world top players use in cross-country skiing classic races, in what situations, in what proportions. We are carrying out this research with our above belief.

I would like to make this research paper that proves the possibility of using one new high-precision GNSS sensor without any sensors to detect techniques in cross-country skiing. Therefore, we are trying to submit to Sensors.

Of course, depending on the performance level of skiers, the percentage of techniques used may vary. Originally, we recognize that the most important aspect of sports science is to measure a lot of skiers and examine the difference in the technique used depending on the performance level. It has been considered that high level skiers are likely to use the double pole technique a lot. However, there is no such data in the world what percentage of the race time is actually used for each technique and how it depends on the performance level. Therefore, it is very valuable to be able to show the usage ratio of each technique of the world's top class skier with high accuracy even if one subject.

This measurement takes a great deal of time and energy. It will still take time for many skiers to analyze. The ultimate goal is to create an algorithm that can automatically discriminate the technique used, but it will take more time to get there. Therefore, as the first step, we thought this research is necessary to prove that the GNSS sensor used in this study can be useful for technique analysis even if the subject was only one.

To that end, we wanted to use as much subjects as possible in our future research.

Point 2

I understood your opinions well.

Maybe my paper was not written well.

The data processing procedure was as follows.

Video data was used as a reference.

2) Next, the number, speed, and cycle time of each technique were measured using GNSS raw data and GNSS raw data extracted by moving average data.

The moving average data was used to create a course profile and to extract the vertical movement of the head from the raw data. The technique count is calculated from raw data and the data obtained by extracting only the vertical movement of the head from the raw data.

3) By using these data, the ratio of GNSS data based on video data was calculated for each technique.

Therefore, exceeding 100% indicates that the number of times counted by GNSS was higher than the number calculated by video. On the other hand, being below 100% means that the number of times counted by GNSS was less than the number counted by video.

I agree with that, as you suggested, we did not calculate match ratio.

I thought the most suitable word for our calculation was % Difference, not match ratio. Therefore, I changed the word “match ration” to “%Difference” in the sentence and table.

Point 3

The discussion presented in section 4 and the conclusions in section 5 are biased, and incorrect.

If my perception is wrong, please point it out again.

We revised section 4 and conclusion statements in Discussion.

Round 2

Reviewer 1 Report

Accept in present form

Reviewer 2 Report

My suggestions are as follows:
1. I recommend a professional major English revision (e.g., using the services provided by the Sensors journal). Attached to this email is the PDF version of the manuscript with some of these needed corrections noted along this document.
2. Concerning the use of “%Difference”, I believe that the most appropriate term will “%Match”, because the autors are measuring how “coincident” is your measure with the “video measure”. I recommend the replacement of “%Difference” by “%Match”.
